# Tablet of *Ximenia Americana* L. Developed from Mucoadhesive Polymers for Future Use in Oral Treatment of Fungal Infections

**DOI:** 10.3390/polym11020379

**Published:** 2019-02-20

**Authors:** Lucas Almeida, João Augusto Oshiro Júnior, Milena Silva, Fernanda Nóbrega, Jéssica Andrade, Widson Santos, Angélica Ribeiro, Marta Conceição, Germano Veras, Ana Cláudia Medeiros

**Affiliations:** 1Laboratório de Desenvolvimento e Ensaios de Medicamentos, Centro de Ciências Biológicas e da Saúde, Universidade Estadual da Paraíba, R. Baraúnas, 351, Cidade Universitária, 58429-500, Campina Grande, Paraíba, Brasil; lucasdealmeida2112@gmail.com (L.A.); nogueiracmilena@gmail.com (M.S.); fernandapontesn@gmail.com (F.N.); jessicafarmacia2017@gmail.com (J.A.); widsonmichael@gmail.com (W.S.); angelyca.p07@hotmail.com (A.R.); 2Centro de Tecnologia e Desenvolvimento Regional, Universidade Federal da Paraíba, Av. dos Escoteiros, s/n, Mangabeira VII, 58055-000, João Pessoa, Paraíba, Brasil; martamaria8@yahoo.com.br; 3Laboratório de Química Analítica e Quimiometria, Centro de Ciências Biológicas e da Saúde, Universidade Estadual da Paraíba, R. Baraúnas, 351, Cidade Universitária, 58429-500, Campina Grande, Paraíba, Brasil; germano@uepb.edu.br

**Keywords:** *Ximenia americana* L., compatibility, hydrophilic polymers, mucoadhesive tablets

## Abstract

The use of biocompatible polymers such as Hydroxypropylmethylcellulose (HPMC), Hydroxyethylcellulose (HEC), Carboxymethylcellulose (CMC), and Carbopol in solid formulations results in mucoadhesive systems capable of promoting the prolonged and localized release of Active Pharmaceutical Ingredients (APIs). This strategy represents a technological innovation that can be applied to improving the treatment of oral infections, such as oral candidiasis. Therefore, the aim of this study was to develop a tablet of *Ximenia americana* L. from mucoadhesive polymers for use in the treatment of oral candidiasis. An *X. americana* extract (MIC of 125 μg·mL^−1^) was obtained by turbolysis at 50% of ethanol, a level that demonstrated activity against *Candida albicans*. Differential Thermal Analysis and Fourier Transform Infrared Spectroscopy techniques allowed the choice of HPMC as a mucoadhesive agent, besides polyvinylpyrrolidone, magnesium stearate, and mannitol to integrate the formulation of *X. americana*. These excipients were granulated with an ethanolic solution 70% *v*/*v* at PVP 5%, and a mucoadhesive tablet was obtained by compression. Finally, mucoadhesive strength was evaluated, and the results demonstrated good mucoadhesive forces in mucin disk and pig buccal mucosa. Therefore, the study allowed a new alternative to be developed for the treatment of buccal candidiasis, one which overcomes the inconveniences of common treatments, costs little, and facilitates patients’ adhesion.

## 1. Introduction

Oral candidiasis is a fungal infection that affects the superficial epithelium of the oral mucosa, most often caused by a disordered growth of *Candida*. Even though it is not a lethal disease, oral candidiasis should be avoided to prevent the invasion of other tissues and a subsequent development of systemic infection. Although infrequent, disseminated candidiasis has a mortality rate of 47% [1,2].

Treatment of oral candidiasis can be topical or oral and commonly involves representatives of the azole class, such as fluconazole and ketoconazole. However, these drugs have disadvantages, such as hepatic side effects, a relatively high disease recurrence rate, and fungal resistance. Compared to synthetic antimicrobial drugs, medicinal plants have the advantages of not causing serious side effects and of not causing microbial resistance to treatment. Studies also show the modulating effect of plant extracts on the microbial resistance of synthetic Pharmaceutical Active Ingredients (APIs) against multi-resistant strains. The use of plants for therapeutic purposes is a popular practice supported by the World Health Organization (WHO), and it should be seen as an alternative in combatting oral candidiasis [3,4,5].

For this reason, research has investigated the pharmacological activities of plant species, particularly *Ximenia americana*. In folk medicine, *X. americana* leaves are used to treat infections in wounds and promote healing effects; its bark tea is used to combat hepatitis and malaria. The bark decoction is used as an antiseptic and a cicatrizant in wounds and snake bites. Due to its antiseptic activity, the seed oil is commonly applied cosmetically, in addition to having a high nutritional value. Several studies have already indicated the antimicrobial, anti-inflammatory, hepatoprotective, hypoglycemic, and antioxidant effects of *X. americana* [6,7,8,9,10,11,12].

Phytochemical investigations using aqueous, ethanolic, and hydroalcoholic extracts of *X. americana* demonstrate the presence of condensed tannins, hydrolysable tannins, saponins, polyphenols, and flavonoids, which are responsible for various biological activities [13,14,15]. Santana et al. (2018) characterized a hydroalcoholic extract of *X. americana*, and their results showed that it was possible to identify and quantify gallic acid as a chemical marker of *X. americana*. In addition, they developed *X. americana* tablets to combat antifungal and bacterial infections, highlighting the pharmacological potential of the species and its phytochemical components [14,16].

Thus, given its wide geographical distribution and its biological potential, developing pharmaceutical formulations with *X. americana* is an innovative and promising proposal [8,15,17,18]. Within this context, the use of novel release forms as mucoadhesive tablets has resulted in a number of advantages over conventional formulations for treating oral infections, such as a longer residence time in the mucosa (saliva and mucosal movements remove conventional formulations quickly), increased permeability of the drug (against degrading agents present in the biological environment), and avoidance of the first-pass effect [19,20,21].

However, developing mucoadhesive tablets requires the choosing of suitable polymers to compose the formulation. The polymers must be capable of adhering to the mucous membranes of the human body, providing temporary retention at the action site, and increasing drug efficacy and adherence to treatment. In pharmaceutical forms, this property is widely used to develop polymeric release systems for oral, nasal, ocular, and vaginal use [22].

The hydrophilic and biocompatible polymers hydroxyethylcellulose (HEC), hydroxypropyl methylcellulose (HPMC), carboxymethylcellulose (CMC), and carbopol have an excellent mucoadhesive capacity. In addition, such polymers are characterized by a molecular matrix that increases the size of their pores according to their swelling ability, promoting a controlled release of the drug through diffusion [19,23,24,25].

Thus, the first step in developing any herbal pharmaceutical form requires detailed compatibility studies aimed at determining the most suitable adjuvants in order to compose the pharmaceutical formulation. These studies intend to characterize physical and chemical incompatibilities that may occur between drugs and pharmaceutical adjuvants, and they represent an important tool in producing a formulation with adequate stability and safety characteristics [26,27].

Therefore, this study aimed to evaluate the antifungal activity of *X. Americana* and to develop a mucoadhesive pharmaceutical form from a study of compatibility between the Plant Active Pharmaceutical Ingredient (PAPI) and pharmaceutical excipients for the treatment of oral candidiasis.

## 2. Materials and Methods

### 2.1. Materials

Carbopol (*M*_w_ = 940.00 g·mol^−1^), colloidal silicon dioxide (*M*_w_ = 60.08 g·mol^−1^), magnesium stearate (*M*_w_ = 591.24 g·mol^−1^), and polyvinylpyrrolidone K-30 (*M*_w_ = 50.00 g·mol^−1^) were purchased from Henrifarma Produtos Químicos e Farmacêuticos Ltda. (Cambuci, Brazil). Aspartame (*M*_w_ = 294.30 g·mol^−1^) was purshased from Mapric Produtos Farmacocosméticos Ltda. (São Paulo, Brazil); sodium saccharin (*M*_w_ = 205.16 g·mol^−1^) from Via Farma—Distribuidora de medicamentos Ltda. (São Paulo, Brazil); fructose (*M*_w_ = 180.16 g·mol^−1^) from Rem Produtos Farmacêuticos Ltda. (Campina Grande, Brasil); sodium carboxymethylcellulose (*M*_w_ = 262.19 g·mol^−1^) from Pharmachemical Comércio e Produtos Farmacêuticos Ltda. (São Caetano do Sul, Brazil); hydroxyethylcellulose (*M*_w_ = 806.94 g·mol^−1^) from Apsen Farmaceutica S/A (São Paulo, Brazil); Hydroxypropylmethylcellulose (*M*_w_ = 1261.45 g mol^−1^) from Unna Derme Comércio de Produtos Farmacêuticos Ltda. (Campina Grande, Brazil); talc (*M*_w_ = 379.26 g·mol^−1^) from Sintética Distribuidora Química Farmacêutica Ltda. (Capivari, Brazil); mannitol (*M*_w_ = 182.17 g·mol^−1^) from Allchem Química Indústria e Coméricio Ltda. (Rio Grande, Brazil); and lactose monohydrate (*M*_w_ = 360.31 g·mol^−1^) from Galena Química e Farmacêutica Ltda. (Campinas, Brazil).

### 2.2. Plant Material

Barks of *X. americana* L. were collected from the semiarid region of Paraíba State, Brazil. Exsiccata was prepared and indentified at the Professor Jayme Coelho de Morais Herbarium, located at the Federal University of Paraíba (Areia city, Paraíba), under the voucher number EAN-100493. Raw barks were dried in an air-forced oven operating at 40 °C and then powdered using a knife mill.

### 2.3. Obtaining the X. Americana Extracts

The extracts were prepared by three different extractive methods: turbolysis or turbo-extraction, maceration, and ultrasonic waves. Hydroethanolic solutions were prepared in varying proportions (50:50, 30:70 and 10:90, *v*/*v*) containing 20% (*w*/*v*) vegetable drug. In the turbolysis, the extract was subjected to high shear agitation using Ultra-turrax^®^ apparatus (IKA, Campinas, Brazil) at 6000 rpm for 15 min under an ice bath for temperature maintenance. The maceration was carried out in a static manner, without solvent renovation, conditioning the extractive solutions in amber glass for seven days under occasional stirring. Ultrasonic wave extraction was performed in an ultrasonic washer (Ultrasonic Cleaner—UNIQUE, Indaiatuba, Brazil) in a water bath at 40 °C for a period of 60 minutes.

All liquid extracts were filtered, concentrated under reduced pressure on a rotary evaporator, and then dried in an air oven at 40 °C. The dried extracts were stored in hermetically sealed vials under refrigeration (± 5 °C) until further analysis.

### 2.4. Evaluation of Antifungal Activity

The antifungal activity was evaluated in vitro by the broth microdilution method, determining the Minimum Inhibitory Concentration (MIC) of each extract. Standard American Type Culture Collection (ATCC) strains of Candida albicans (18804) were used. The microbial suspension was standardized in a UV–VIS (UVmini-1240—Shimadzu, Kyoto, Japan) spectrophotometer at a wavelength of 530 nm to contain the equivalent of 10^6^ CFU mL^−1^. The extracts were solubilized in 10% DMSO. Nystatin was used as a positive control [28].

### 2.5. Binary Mixtures

Physical mixtures of the dry extract (AMX) and pharmaceutical excipients were prepared by geometric dilution in the proportion 1:1, 1:2, and 2:1 (AMX: excipient *w*/*w*). Functional categories of each selected excipient are shown in Appendix A (Table A1). The compatibility study was conducted by analyzing the binary mixtures by Differential Thermal Analysis (DTA) and Fourier Transform Infrared Spectroscopy (FTIR) [29].

### 2.6. Thermal Analysis

The DTA curves of AMX, binary mixtures, and pharmaceutical excipients were obtained on a simultaneous DTA/TGA DTG-60 (Shimadzu, Kyoto, Japan) thermal analyzer using an aluminum sample holder containing a 2 ± 0.1 mg sample under an atmosphere of nitrogen with a flow of 50 mL·min^−^^1^. The samples were subjected to heating in a temperature range of 25 to 400 °C in programming of 10 °C·min^−^^1^. For calibration of the equipment, Indium (melting point 156.6 °C) was used as the standard.

Thermogravimetry (TG) was only used for the characterization of the extract. Thus, the thermogravimetric curve of the AMX was obtained from a simultaneous thermal analyzer previously mentioned for DTA analysis, using an alumina sample holder containing 8.0 ± 0.5 mg of sample, under nitrogen atmosphere at a flow rate of 50.0 mL·min^−^^1^ as the purge gas. The sample was conditioned at a temperature range of 25–900 °C at a heating rate of 10 °C·min^−^^1^. The data were analyzed using the TA60-WS software (Shimadzu, Kyoto, Japan).

### 2.7. FTIR

Absorption spectra in the infrared region were obtained from the Shimadzu Spectrophotometer, IRPrestige model (Shimadzu, Kyoto, Japan), using KBr pellets, at the range of 4000–400 cm^−^^1^. Data were analyzed using Origin^®^ software, version 8.0 (OriginLab Corporation, Northampton, MA, USA).

### 2.8. Formulation Development

From the compatibility study performed by DTA and FTIR, the pharmaceutical excipients that were compatible with the AMX plant extract were selected to form the formulation. Based on the chosen excipients, different formulations were proposed, following the concentration limits recommended for each component [29]. The proposed formulations were then monitored for their compressibility and flow properties, and the best performance formulation was chosen for tableting through direct compression, using a Lemaq Monopress LM-1 compressor (Lemaq, Diadema, Brazil) [30,31,32].

### 2.9. Mucoadhesion

The mucoadhesive strength of the formulation was analyzed using a TAXT plus texture analyzer (Stable Micro Systems^®^, Surrey, UK). Mucin disks or pig buccal mucosa were used for mucoadhesion analysis. Initially, the mucin disc was prepared by compressing raw swine mucin (250 mg) moistened with 50 μL of 8% (*w*/*w*) mucin dispersion and using a tablet compressor with a diameter of 123 mm, and the pig buccal mucosa was obtained from a local slaughterhouse. First, 50 µL of artificial saliva was applied to the surface of the mucin disk before the experiment and the mucosa was immersed in human saliva to simulate the buccal environment for 30 s. Then, the mucin disks or pig buccal mucosa were taped horizontally in a cylindrical probe of the texturometer with double-sided tape to keep them static. After that, the tablet was adhered to the surface of the lower acrylic plate and was placed below the cylindrical probe, thereby triggering the lowering of the cylindrical test at a rate of 1 mm/sec until the mucin disc reached the tablet. The cylindrical probe was kept in contact with no force applied for 60 s to ensure intimate contact between the tablet and the mucin disc. After this time, the test was ended at a speed of 1 mm/sec [33,34]. During the experiment, a force-time curve was recorded through the Expert Texture Exponent 32 software (version, Stable Micro Systems, Surrey, UK) and the area under the force-distance curve during the withdrawing phase and peak adhesion was calculated as the work of adhesion (Wad). This process was replicated five times at 37 ± 1 °C.

## 3. Results and Discussion

### 3.1. Evaluation of Antifungal Activity

The results obtained for determining the antifungal activity of the *X. americana* rotavapor extracts are described in Table 1.

The extracts evaluated in this study showed inhibitory activity in a concentration range between 125 and 250 μg·mL^−1^. The MIC of 125 μg·mL^−1^ was presented by extracts of 70% and 90% ethanol for maceration, 50% and 70% for turbolysis, and 50% for ultrasound. Previous research has demonstrated the antifungal activity of a chloroform extract of *X. americana* against *C. albicans*, as well as the absence of activity from methanolic and aqueous extracts; we also found another study that reported weak activity of the methanolic and aqueous extracts of the stem bark [35,36].

In contrast, a study comparing the action of 31 plant species against *C. albicans* showed that only six inhibited it, and the methanolic extract of *X. americana* showed one of the lowest IC_50_ values obtained in the study (8.12 μg·mL^−1^). Previous studies also reported the activity of other species of the genus Ximenia against *C. albicans,* indicating the potential of these species in the treatment of oral candidiasis, which had already been reported in ethnopharmacological studies [37,38,39].

Studies suggest that most of the secondary metabolites present in plant species have antifungal activities, and the synergism of the actions of these compounds is an effective alternative in treating fungal infections. Previous research has suggested that phytocomposites can inhibit fungal cell wall formation and cause cell membrane rupture, fungal mitochondria dysfunction, inhibition of cell division, inhibition of RNA/DNA and/or protein synthesis, and inhibition of efflux pumps [40,41,42].

With the similar values of MIC in mind, we chose the AMX extract obtained by turbolysis at 50% of EtOH to perform the PAPI-excipient compatibility study because it is a cheaper, faster method and an extractive solution with lower alcohol. The use of ethanol as a solvent is a strategy that follows green chemistry principles, since it is more appropriate biologically and environmentally [33].

### 3.2. Thermal Characterization of AMX

The DTA curve of AMX (Figure 1a) shows the presence of an endothermic peak around 98.67 °C (Δ*H* = 283.69 J·g^−^^1^), certainly related to the loss of volatile compounds, ethanol, and water in the sample. This event at the TG curve (Figure 1b) is related to a mass loss of 9.56% of the sample. At the TG curve, in reference to the decomposition of organic compounds, a loss of mass equivalent to 51.94% was observed starting at 211 °C. The residue that formed at the end of the heating corresponded to 38.5% of the total mass analyzed.

### 3.3. Compatibility Study

#### 3.3.1. DTA

Thermoanalytic techniques, such as DTA, are widely used to detect incompatibilities in a short period of time, with the curve of the mixture being a result of the individual curves of each substance analyzed. The suppression, disappearance, appearance, or displacement of thermal events, as well as variation in the expected values of enthalpy, should be considered as possible incompatibility. These parameters were used to evaluate the binary mixtures described below [43].

The DTA curves of the binary mixtures of the AMX plant extract and the pharmaceutical excipients are shown in Figure 2, Figure 3 and Figure 4, and the data are described in Appendix A (Table A2). Most of the mixtures showed significant variations of their thermal profiles. The aspartame thermoanalytical curves (Figure 2a) recorded a shift of the melting temperature of the pharmaceutical excipient (249.17 °C) at all mixing ratios—1:1 (235.55 °C), 1:2 (237.58 °C), and 2:1 (234.78 °C)—suggestive of physical incompatibility. Variations of the energy involved in these thermal events were also observed and predicted according to the proportion of the mixture [29].

Signs of AMX-pharmaceutical excipient incompatibility were observed with carbopol (Figure 2b), in the curves of which a drastic reduction of the energy involved in the stages of the decomposition of CBP was observed, leading to the disappearance of these peaks. This can be considered as a physical incompatibility, despite the presence of AMX characteristics in the thermal profiles of the mixtures [44].

The thermal curves of the mixtures with the carboxymethylcellulose (Figure 2c) and the colloidal silicon dioxide (Figure 2d) demonstrated the preservation of the characteristics of both AMX and the pharmaceutical excipients, thus indicating no apparent physical incompatibility between them.

According to Figure 3, physical incompatibility was also possible in the AMX-fructose mixtures (Figure 3a). It shows displacement of melting temperatures (107.92 °C) and decomposition (210.13/279.24 °C) of the FRU at the ratio 1:1 (126.09/195.05/222.28 °C), anticipation of thermal events at 1:2 (101.92/149.76/237.24 °C), and the junction of all events at 2:1, resulting in a single high-intensity endothermic peak (Δ*H* = −294.63 J·g^−1^) at 121.37 °C [29].

For the AMX-hydroxyethylcellulose mixture (Figure 3b), suppression of the second HEC decomposition event (364.60 °C/Δ*H* = 10.15 J·g^−1^) was observed at 1:1 and 1:2, and a temperature delay of this event was seen at 2:1 (383.22 °C) with suppression of the first HEC decomposition event (343.28 °C), suggesting possible incompatibility, despite the presence of thermal characteristics of the AMX in the mixtures. These characteristics were also represented in the binary mixtures with hydroxypropyl methylcellulose (Figure 3c), in which no apparent alterations were observed in the thermal profiles that characterize incompatibility with the AMX.

Analyzing the AMX-lactose curves (Figure 3d), we were able to verify some evidence of physical incompatibility. Thermal analysis of the 1:1 ratio mixture showed a change in the temperature and melting energy of the LAC in the mixture, from 219.34°C (Δ*H* = −226.34 J·g^−1^) to 207.32°C (Δ*H* = −4.14 J·g^−1^), as well as the disappearance of the second decomposition event of this excipient (304.85 °C/Δ*H* = −137.36 J·g^−1^) [29]. Displacement at this temperature was also found in ratios 1:2 (211.31 °C) and 2:1 (210.83 °C), in addition to the suppression of decomposition events of the pharmaceutical excipient (241.72/304.85 °C).

Two endothermic events characterized the thermal profile of mannitol (Figure 3e): melting (169.89 °C/Δ*H* = −450.53 J·g^−1^) and decomposition (321.83 °C/Δ*H* = −812.74 J·g^−1^). The melting point found is close to the one described in the literature, at 166–168 °C [45]. Signals of AMX-MAN incompatibility were verified in the binary mixtures, with displacement of these events at 1:2 (165.63/304.15 °C) and 2:1 (162.94/325.73 °C). At the 1:1 ratio, the anticipated melting temperature was observed, along with a drastic energy decrease (163.10 °C/Δ*H* = −50.80 J·g^−1^) and the disappearance of the peak decomposition of the pharmaceutical excipient (321.83 °C).

For magnesium stearate (Figure 4a), the endothermic AMX moisture loss peaks (98.67 °C/Δ*H* = −283.69 J·g^−1^) and melting point (129.00 °C/Δ*H* = 255.25 J g^−1^) melted, resulting in a single low energy peak in the 1:1 ratio mix (127.00 °C/Δ*H* = −71.44 J g^−1^). Commercial samples of magnesium stearate have a melting range of 117-150 °C [29]. Still in the ratio 1:1, the magnesium stearate decomposition event (319.75 °C/Δ*H* = −379.99 J·g^−1^) was seen to be suppressed. In addition, displacement of this temperature was observed at the 1:2 mixture (341.03 °C/Δ*H* = −204.32 J·g^−1^), indicating physical incompatibility in the AMX-MST mix.

In the literature, a possible explanation put forth for incompatibility with magnesium stearate is its chemical nature, a mixture of organic salts formed by magnesium cations and anions from different fatty acids that can generate chemical reactions with the active compounds by forming new products of degradation [46].

Saccharin also showed evidence of physical incompatibility with AMX (Figure 4c). Saccharin salts have a melting temperature above 300 °C; the melting temperatures determined in the mixtures still remained close to that of the excipient (365.15 °C/Δ*H* = −210 J·g^−1^), but the energy spent in these thermal events was significantly reduced at 1:1 (356. 32 °C/Δ*H* = −26.91 J·g^−1^), 1:2 (357.94 °C/Δ*H* = −65.39 J·g^−1^), and 2:1 (356.49 ° C/Δ*H* = −18.02 J·g^−1^) [47].

Polyvinylpyrrolidone K-30 (Figure 4b) and talc (Figure 4d) maintained their thermal characteristics in a mixture with AMX, demonstrating that no changes occurred that were not predicted.

In summary, according to the results of this study, CSD, HPMC, CMC, PVP, and TAL were the pharmaceutical excipients that showed no evidence of physical incompatibility with the AMX. The binary mixtures were then submitted to FTIR for confirmation of possible incompatibilities.

PAPI-excipient compatibility studies are important tools that predict possible reactions between the formulation components that may occur during the storage period under storage conditions. The combination of thermal techniques (such as DTA) and non-thermal techniques (such as FTIR) is successful in identifying and confirming incompatibilities. Since these techniques present different work fundamentals and vary the analysis time, amount of sample, and mechanical and/or thermal stress employed, the results obtained are complementary and provide different conclusions. However, in the absence of evidence of interaction, compatibility should be confirmed by FTIR [43,48].

#### 3.3.2. FTIR

The FTIR spectra of the AMX extract and their binary mixtures are set forth in Figure 5, Figure 6 and Figure 7. According to the FTIR spectrum obtained from the AMX (Figure 5), a wide and intense absorption band is visible between 3800 and 3000 cm^−^^1^, suggestive of O–H stretching. These hydroxyls may be related to the phenolic compounds and moisture content present in the extract, since it is an amorphous and hygroscopic sample. In addition, the presence of organic compounds in the extract was characterized in the spectrum by the occurrence of a low-intensity peak in the range between 3000 and 2850 cm^−^^1^, indicative of C–H stretching. Acute peaks observed between 1600 and 1475 cm^−^^1^ are possibly related to C=C bonds of aromatics. Near this region, characteristic peaks of C–H were observed for folding between 1450 and 1375 cm^−^^1^, as well as peaks suggestive of C–O stretching in the range of 1300 to 1000 cm^−^^1^. Between 788 and 674 cm^−^^1^, narrow peaks of a low intensity were visualized, probably related to C–H bonds in substituted aromatics. These chemical bonds refer to a variety of functional groups—such as ethers, esters, and carboxylic acids—that accompany the chemical composition of flavonoids, tannins, anthraquinones, and other secondary metabolites present in the extract [16].

Indices of AMX-pharmaceutical excipient chemical incompatibility were evaluated, taking into account the emergence of new bands, as well as changes in the absorption ranges and/or intensity of the characteristic peaks of the extract and excipient in the binary mixtures [27,46].

The spectra of the binary mixtures of AMX with aspartame (Figure 5a) did not show any apparent changes in the spectroscopic profile of the plant extract; however, the characteristic peaks of the pharmaceutical excipient were predominantly observed in all the analyzed spectra, even at a ratio with a higher proportion of the extract (2:1). The peaks related to the AMX were not very pronounced throughout the spectra, due to the overlap of absorption bands of the pharmaceutical ingredient. The overlapping bands are not considered a parameter that is indicative of chemical incompatibility [49]. The decrease in peak intensity, observed in the 1:2 and 2:1 mixtures, may be attributed to possible changes in the geometric mixture of the components, since this reduction occurred on the spectrum as a whole. Therefore, there was no evidence of chemical incompatibility between AMX and ASP.

With respect to the mixtures of AMX and carbopol (Figure 5b), representative absorption bands of the plant extract could be visualized in all mixing proportions, with overlapping rare peaks. As expected, the intensity of the bands was also found to vary between the mixtures, according to the proportion. The same behavior was observed in mixtures of AMX and carboxymethylcellulose (Figure 5c). Therefore, it is clear that no evidence shows that AMX interacted chemically with these pharmaceutical excipients.

The binary mixtures of AMX-colloidal silicon dioxide (Figure 5d) showed spectra consistent with the characteristics of each of the components of the mixture. However, the intensity of the peaks related to the extract was reduced. This can be explained by the presence of a broad and intense band at 1080 cm^−^^1^, suggestive of Si–O–Si bonding of the siloxane groups of the pharmaceutical ingredient, which promotes overlapping and attenuation of the other spectral bands. A chemical compatibility can be seen between AMX and CSD.

Indices of chemical compatibility were also determined in spectra of the AMX mixtures with fructose (Figure 6a), hydroxyethylcellulose (Figure 6b), hydroxypropyl methylcellulose (Figure 6c), lactose (Figure 6d), and mannitol (Figure 6e). The obtained spectra resulted from a sum of characteristics of the extract and the pharmaceutical ingredient evaluated.

For mixtures with FRU (Figure 6a), the intensity of the AMX peaks was shown to be positively influenced by a greater amount of extract in the mixture (2:1), as predicted. In a similar way, the excipient at 1:2 was verified, in which the intensity of the peaks related to FRU increases. In mixtures with HEC (Figure 6b), a different behavior was noted. The total peaks resulting from the mixture are reproducible between the different proportions, and their intensities increase (compared to 1:1 mixture) when the amount of the extract (2:1) or the excipient (1:2) is increased; in other words, 1:2 and 2:1 present similar bands of intensity, showing that both the AMX and the pharmaceutical excipient have an equal influence on the spectrum of the mixture.

The spectra of the AMX-hydroxypropylmethylcellulose mixture (Figure 6c) expressed opposite-than-expected effects with the increasing proportions of extract and excipient in the binary mixtures: the 1:1 proportion mixture had higher peak intensities than those from mixtures with higher amounts of excipient (1:2) and extract (2:1). Band fusion may have occurred at the regions of absorption common to the components of the 1:1 mixture, which could not be seen at the other proportions.

A similar profile to that observed in the HPMC mixture was repeated for lactose (Figure 6d), mannitol (Figure 6e), polyvinylpyrrolidone K-30 (Figure 7b), saccharin (Figure 7c), and talc (Figure 7d) in mixtures with AMX. Generally, the spectra retained the identity of the AMX, as well as those of the pharmaceutical excipients; no displacement of the characteristic absorption regions of the mixtures’ components was observed. Some bands related to the extract were superimposed because the region of spectral absorption was with the same as that for the excipients; however, the intensity of the absorption bands was verified between the mixtures within what was expected or could be explained. Thus, no evidence was shown for chemical incompatibility between these pharmaceutical excipients and AMX.

For mixtures with magnesium stearate (Figure 7a), the intensity of the AMX and excipient peaks were shown to be directly related to the component concentration in the mixtures: AMX characteristic absorption bands were most evident at the 2:1 proportion mixture, and characteristic peaks of the MST were more intense at the 1:2 proportion mixture, as expected.

It is important to note that for the AMX-talc mixtures, the peak intensity of the extract significantly changed, as observed for the AMX-colloidal silicon dioxide mixtures. Once again, this phenomenon can be attributed to the siloxane groups in the pharmaceutical excipient, the presence of which causes an intense absorption at 1080 cm^−^^1^, thus promoting attenuation of the other spectral bands [50].

Therefore, the evaluation of the binary AMX-pharmaceutical excipient mixtures by FTIR made it possible to determine that all the excipients studied did not show evidence of chemical incompatibility with the AMX plant extract.

### 3.4. Mucoadhesive Tablet Development

Based on the results obtained in the compatibility studies with DTA and FTIR, the pharmaceutical excipients HPMC, MAN, LAC, MST, and PVP were chosen for the formulation. Different formulations were proposed, fixing the concentrations of MAN, MST, PVP, and AMX and varying the proportions of HPMC and LAC, according to the manual of pharmaceutical inputs [29].

However, unsatisfactory flow was observed when LAC concentrations were added and increased. Therefore, LAC was removed, since the diluent function was already exerted by MAN. The AMX concentration was determined based on a microbiological potency study carried out by our research group. Table 2 shows the composition of the final formulations.

The final formulation showed an angle of repose between 31° and 35°, characterized as a good flow. However, the Carr Index (CI) and Hausner’s Factor (HF) were classified as very poor flow (CI between 32–37%, HF between 1.46–1.59). Thus, it was necessary to granulate the powder using an ethanolic solution (70% (*v*/*v*) at 5% PVP).

Table 3 shows the characterization data of flow to granulation formulation. The results reveal adequate flow, angle of repose < 30°, CI 12%, and HF 1.13, therefore exhibiting suitability for direct compression [31].

### 3.5. Mucoadhesion

Texture Profile Analysis (TPA) is a technique that allows the determination of the mucoadhesive strength (negative area of the force-time curve per unit time required to detach the tablet from the substrate during the first compression cycle) between the tablet and the mucin disk or pig buccal mucosa. The values of a commercial formulation adhesion call (Triancinolone orabase) were used to verify if the mucoadhesive tablet had suitable interactive properties [51].

Figure 8 shows the work of mucoadhesion values (Wad) for the mucoadhesive tablet and the commercial cream formulation.

When the value of the mucoadhesion between the tablet (0.177 ± 0.011 N) and the adherent commercial formulation (0.1972 ± 0.023 N) using a mucin disk is compared, the value of the work of mucoadhesion displays no statistical difference. Furthermore, we can verify that both materials analyzed in pig buccal mucosa presented values statistically lower than those presented with the mucin discs. However, when comparing the mucoadhesive values the between tablet (0.022 ± 0.008 N) and commercial formulation (0.128 ± 0.05 N), the tablet has statistically lower adhesion than the commercial formulation. Bespite this, all values 55 obtained in this test have higher work of mucoadhesion than values obtained in other studies dealing with mucoadhesive systems that use the same methodology, supporting the proposal of using these materials as mucoadhesive systems [52,53,54]. Perioli et al. (2008) demonstrated that an increase of the compression force on tablet increases the mucoadhesion values and tablets with the lowest force (0.57 N) are better than tablets with values above 1.20 N because they cause pain during in vivo application [55].

The obtained results are due to the chemical structure of polymers used in the tablet. This includes a sufficient amount of chemical groups capable of forming hydrogen bonds (–OH, –C=O and –NH2) with the biological substrate and swelling capacity, which enables mucosal variations to be adjusted and helps interpenetration between chains of the mucus and the polymeric chains of the tablet. One of the most predominant concepts that explains the process of mucoadhesion is chemo-adsorption, a subtype of the adsorption theory. According to chemo-adsorption, adhesive bonding occurs through secondary forces, including hydrogen bonds, hydrophobic bonds, Van Der Walls interactions, and electrostatic forces, established between the atoms of the bioadhesive material and those of the muco-glycoprotein network; despite being considered as weak interactions, all these together produce an intense adhesive force [56,57,58].

In this way, the structure of the polymer is an important and influential factor in the mechanism of mucoadhesion. HPMC, a nonionic hydrophilic polymer, has mucoadhesive properties due to the hydroxyl groups present in its structure, making it capable of establishing hydrogen bonds with mucus glycoproteins [59,60]. Hydroxypropyl methylcellulose is often used to produce sustained release matrix tablets due to its biocompatibility, biodegradability, easy handling, low cost, ease of compaction, and high drug-loading capacity. Moreover, and most importantly, the release of the drug promoted by the hydrophilic matrix of this polymer is little affected by the process variables [61,62].

Mucoadhesion may improve the administration of antifungal drugs by allowing release of the active ingredient in a specific target region affected by infection. By establishing a consistent interaction between the polymer and mucus, the increased contact time between the two surfaces promotes increased drug action in situ, providing a greater therapeutic efficacy [63,64,65].

In vivo studies indicate that mucoadhesive tablets may achieve an equal or better therapeutic performance compared to conventional formulations in treating oral candidiasis. Comparing the efficacy and safety of miconazole nitrate tablets and itraconazole capsules in patients diagnosed with oral candidiasis, Yan et al. (2016) found a cure rate of 51.18% for miconazole and 41.76% for itraconazole. They also observed no significant differences between these drugs regarding the safety profile of the patients [66]. Evaluating mucoadhesive tablets of nystatin in healthy volunteers, Llabot et al. (2009) found that the tablets resided in the mucosa of the patients for 4.5 hours. During this period, the concentration of nystatin in the saliva remained higher than the MIC of the drug, according to the behavior observed in vitro [67].

## 4. Conclusions

The results showed that, as complementary analytical techniques, DTA and FTIR were effective in the PAPI-excipient compatibility study. The results also demonstrated that Hydroxypropyl methylcellulose, Mannitol, Magnesium Stearate, and Polyvinylpyrrolidone K-30 are the most suitable pharmaceutical adjuvants in the formulation. In addition, the use of the Hydroxypropyl methylcellulose polymer resulted in a tablet with mucoadhesion values similar to commercial formulations and better than other mucoadhesive systems in the literature. Mucoadhesion force assays are relevant to pharmaceutical formulations, since higher mucoadhesion values may increase the time of action and absorption of the drug, thus improving the clinical performance of the formulation. In this way, we can conclude that this formulation may be used as a mucoadhesive tablet and that incorporating antimicrobial *X. americana* is promising for the treatment of oral infections. The next stage of this work will be to perform in vitro and in vivo experiments in order to prove the efficacy of this new therapeutic approach.

## Figures and Tables

**Figure 1 polymers-11-00379-f001:**
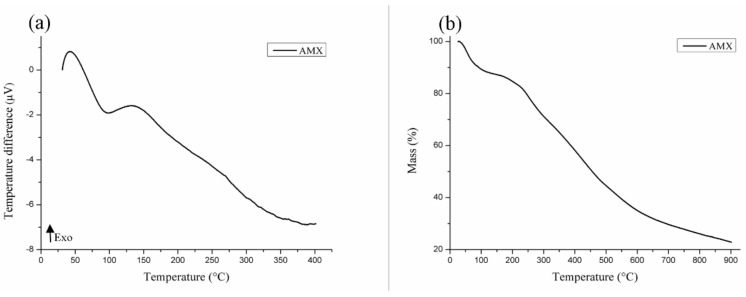
DTA (**a**) and TG (**b**) curves of AMX.

**Figure 2 polymers-11-00379-f002:**
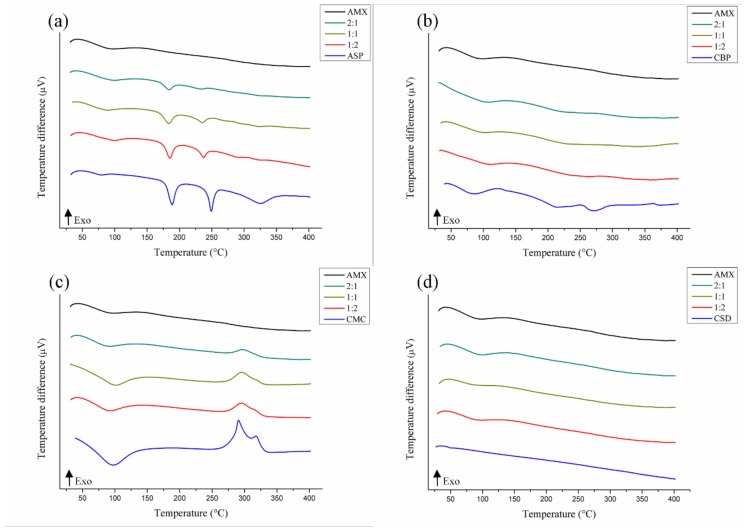
DTA curves of binary mixtures of AMX with pharmaceutical excipients: aspartame (**a**), carbopol (**b**), carboxymethylcellulose (**c**), and colloidal silicon dioxide (**d**).

**Figure 3 polymers-11-00379-f003:**
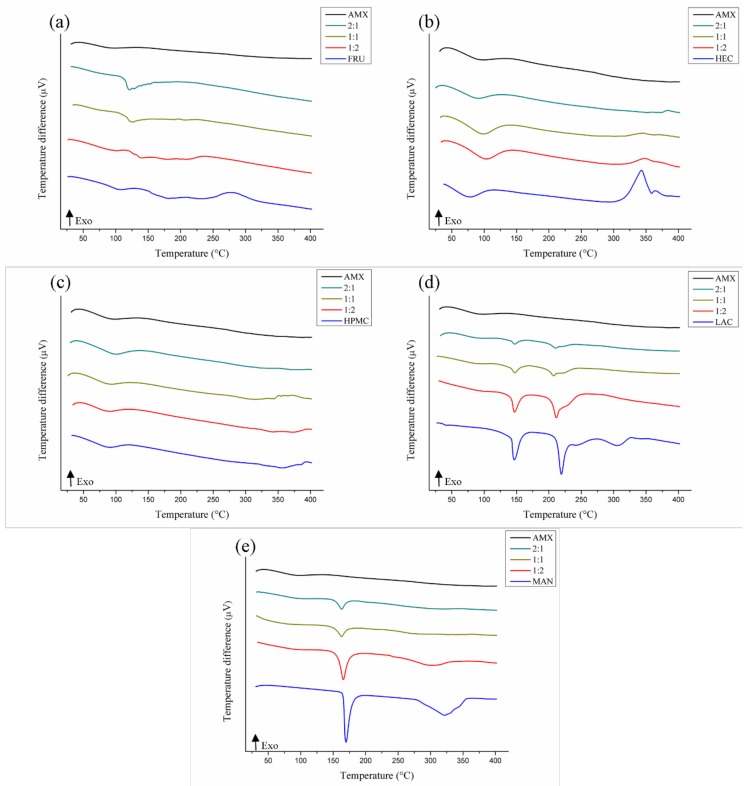
DTA curves of binary mixtures of AMX and pharmaceutical excipients: fructose (**a**), hydroxyethylcellulose (**b**), hydroxypropyl methylcellulose (**c**), lactose (**d**), and mannitol (**e**).

**Figure 4 polymers-11-00379-f004:**
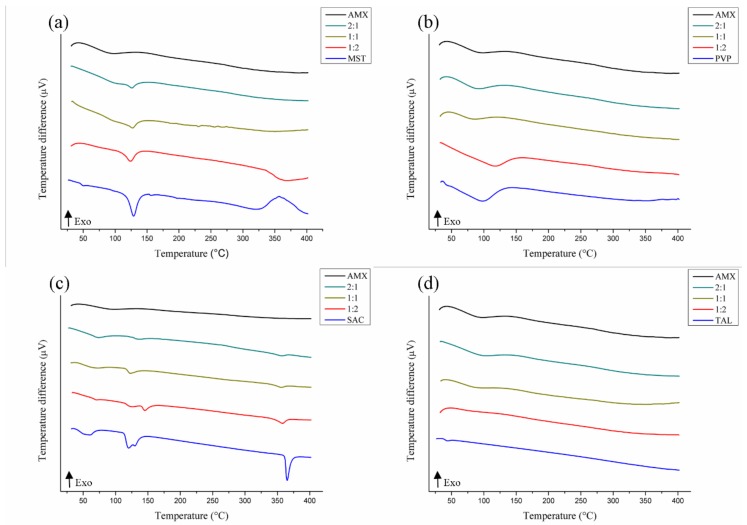
DTA curves of the binary mixtures of AMX and pharmaceutical excipients: magnesium stearate (**a**), polyvinylpyrrolidone K-30 (**b**), sodium saccharin (**c**), and talc (**d**).

**Figure 5 polymers-11-00379-f005:**
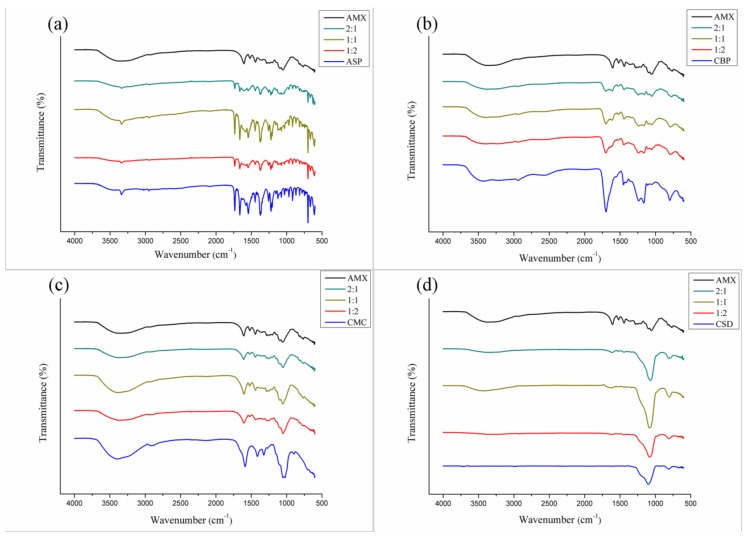
FTIR spectra for AMX and mixtures thereof with pharmaceutical excipients: aspartame (**a**), carbopol (**b**), carboxymethylcellulose (**c**), and colloidal silicon dioxide (**d**).

**Figure 6 polymers-11-00379-f006:**
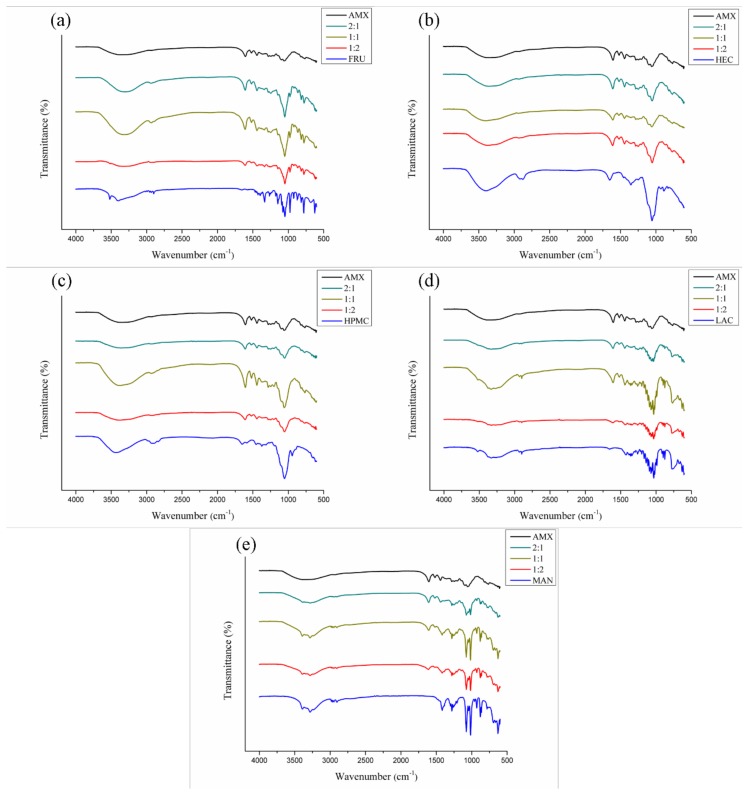
FTIR spectra for binary mixtures of AMX and pharmaceutical excipients: fructose (**a**), hydroxyethylcellulose (**b**) and hydroxypropylmethylcellulose (**c**), lactose (**d**), and mannitol (**e**).

**Figure 7 polymers-11-00379-f007:**
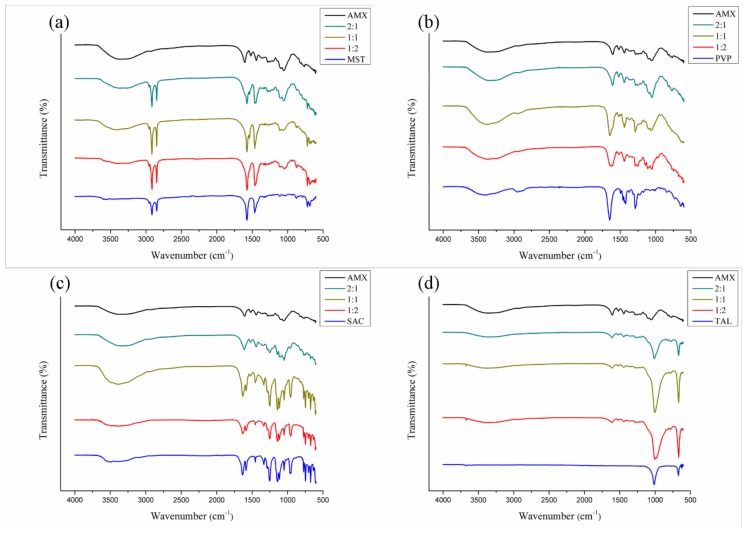
FTIR spectra for binary mixtures of AMX and pharmaceutical excipients: magnesium stearate (**a**), polyvinylpyrrolidone k-30 (**b**), sodium saccharin (**c**), and talc (**d**).

**Figure 8 polymers-11-00379-f008:**
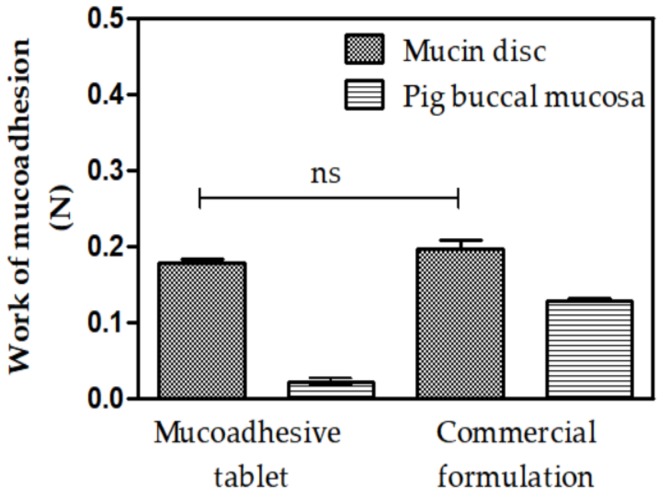
Work of mucoadhesion values for mucoadhesive tablet and commercial mucoadhesive formulations when in contact with mucin discs and pig buccal mucosa. The statistical significance was analyzed using variance analysis via Anova One Way; *p* ≤ 0.05. n.s.: no significant difference. Results are expressed as mean ± SD for *n* = 5.

**Table 1 polymers-11-00379-t001:** Minimum Inhibitory Concentration (MIC) of extracts of *X. Americana.*

		MIC (µg·mL^−1^)		
	EtOH (%)	Maceration	Turbolysis	Ultrasound
*X. americana*	50	250.00	125.00	125.00
	70	125.00	125.00	250.00
	90	125.00	250.00	250.00

**Table 2 polymers-11-00379-t002:** Final formulation of the mucoadhesive tablets.

Excipient	Concentration (%)	Class
HPMC	48.97	Adhesive
MAN	10.00	Diluent/sweetener
MST	2.00	Lubricant
PVP	2.00	Desintegrator
AMX	37.03	PAPI

**Table 3 polymers-11-00379-t003:** Flow characterization data.

Parameters	Formulation	Granulate
Gross density (g·mL^−1^)	0.526	0.550
Density of compaction (g·mL^−1^)	0.769	0.625
Hausner factor	1.460	1.13
Carr index (%)	31.60	12.0
Index of densification (mL)	2.00	1.0
Angle of repose (°)	33.00	0.15
Flow time (s)	4.57	3.12

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
