# Peer review of "Tablet of Ximenia Americana L. Developed from Mucoadhesive Polymers for Future Use in Oral Treatment of Fungal Infections"

_polymers, 2019, doi:10.3390/polym11020379_

Round 1

Reviewer 1 Report

The paper by Almeida et al. focuses on the development of mucoadhesive tablets for the oral treatment of fungal infections. The proposed tablets contain a vegetal drug extracted of the shell of Ximenia Americana, a plant specimen with pharmacological activity. Some hydrophilic polymers were tested as pharmaceutical excipients. The paper evaluates its mucoadhesion as a warranty for improved clinical performance. A compatibility studio between plant active pharmaceutical ingredients (PAFI) and those pharmaceutical excipients was performed by means of differential thermal analysis (DTA) and Fourier Transform Infrared Spectroscopy (FTIR). The results presented are interesting and well detailed, but there are some issues that need to be improved before the manuscript is accepted.

1.    The title of the manuscript is appropriate, but the abstract should be clarified in order to properly reflect the aim of the paper and an appropriate summary.  The indication of trying to achieve high mucoadhesion values should be introduced in the abstract and conveniently justified with support from the literature.  Please, make a more extensive literature review on this subject.

2.     Does the incorporation of X. Americana into other conventional formulations be considered? How it can affect its pharmacological activity? Does the incorporation of other pharmaceutical herbal extracts be considered? Please specify why you decided to choose X. Americana among others. The novelty  and indication should be clearly presented

3.    Mostly, there is no discussion with the results from the literature. For example, the authors write “……Line 251 The overlapping bands are not considered a parameter indicative of chemical incompatibility…” Is this similar to the observation of other authors? The authors should make a short literature review (as they made in line 308, reference 33) on the results subject and show what is new in their research in relation to those already published. So, the novelty should be clearly presented.

4.    The paragraph (Line 322) referring to the comparison of adhesion strength values for the mucoadhesive tablet with a commercial cream formulation is not well introduced. Please, rewrite this part.

5.    Please, homogenize the nomenclature in all captions of figures and tables and revise more carefully the usual typographical mistakes. English must be improved. Some examples: Line 31, 32 (muchoadesive, literaturen…), 75, 79, 92 (exsicata for exsiccata), 112, 113, 128 (componente), 156, 162, and 162 (excipiente), 218, 231, 277, 281 (proportion), 324 (biological substrate), 342 (another), 353 (datas)

Author Response

Thanks to the referee comments. The literature review was realized, and the abstract was modified to better understand and approach the purpose of this study. The change is highlighted in the yellow text and described in "List of Changes in the manuscript text".

Reviewer 2 Report

This work aims to develop a mucoadhesive tablet of Ximenia americana (XA) L. by physical mixing with mucoadhesive polymers and to evaluate its mucoadhesion against a commercial formulation. For this, selected polymers were Hydroxypropylmethylcellulose (HPMC), Hydroxyethylcellulose (HEC), carboxymethylcellulose (CMC) and carbopol. The compatibility, this is the lack of interaction between XA and these polymers and other excipients were evaluated by DTA and FTIR tecnniques.

Taking into account the aim and content of this work, as well as the no significant relevance of the study about employed polymers, I think that it is out of the scope of “polymers” journal and I suggest that it is resubmitted to a more specific journal such as, “Pharmaceutics” or “Pharmaceuticals”. Nevertheless, some points should be improved:

Abstract Lines 30-35 should be rewritten for clarity.

The novelty of the investigation should be highlighted.

The role of all employed excipients should be specified.

What does k-30 mean? For PVP.

A brief summary about reported investigations on X. Americana dosification methods, formulations.., should be added in the introduction.

Authors claim that develop mucoadhesive tablets, however the measurement of a unique sample was carried out. It should be interesting to develop a systematic study about the effect of mucoadhesion of the different studied polymer, combination of tem.

The molecular weight is a crutial parameter on mucoadhesion for a polymer. However they have not been specified. It would have been an interesting point for being studied.

The relation between the employed techniques and the compatibility study should be explained, and obtained results should be interpreted accordingly. How do you observed on a DTA curve that mixing substances are compatibles? This explnation in required for clarity.  Indeed, physico-chemical interactions between components of the formulation through are done by  the comparison of thermal curves of pure substances with the curve obtained from a 1:1 physical mixture and therefore select adequate excipients with suitable compatibility. Why other ratios?

In fact, DSC has been proposed to be a rapid method for evaluating compatibility, it would have been preferred than DTA.

Gradual intensity loss of DTA curves or FTIR is only due to the decrease of the concentration in the sample of the corresponding substance. It does not mean interaction. For this peak appearance or disappearance is required. But the reason of this appearance /disappearance should be specified or hypothesized.

Hypothesis should be made in all the cases about obtained results, what type of interaction? Why the decrease of appearance of a peak? (For example line 282-284)

Curves and FTIR spectrum of  AMX should not be presented separately, it is included in the rest of graphs.

TGA is a very useful technique to study possible interactions, because interactions retard degradation. Decomposition stages are easily observed by TGA (DTG). This technique should have been considered in this work.The election of the final formulation is not properly supported or described.

Work should be completed, content is poor. The kinetic profiles of the degradation of the tablet, the release profile of the X.A, the antibacterial properties, molecular weight effect, ..

Author Response

We would like to thank the Reviewer for their valuable comments concerning our work. All issues raised in the review process are addressed in the Revised Manuscript. The literature review was realized, and the abstract, introduction and results were modified to better understand and approach the purpose of this study.

Reviewer 3 Report

The authors should spell-check the text since not English expressions often occur.

2.1 Plant Material section should be rephrased and clarify the extraction process.

The source and the grade of the selected excipients, including average molecular weights, are missing (See 2.2 Binary mixtures). The authors should add this information.

Author Response

We would like to thank the Reviewer for their valuable comments concerning our work. All issues raised in the review process are addressed in the Revised Manuscript.

Round 2

Reviewer 2 Report

Requested questions about discussion and the incorporation of needed new results, have not been addressed. Taking into account the aim and content of this work, as well as the no significant relevance of the study about employed polymers, certainly polymers are excipients, I think that this paper is out of the scope of “polymers” journal and I strongly suggest that it is resubmitted to a more specific journal such as, “Pharmaceutics” or “Pharmaceuticals”.

Lines  254-256:

The incompatibility between lactose and APIs is ascribed to an chemical reaction aldehyde-amine reaction, not to a physical interaction. Thus, it should have been proven, new chemical bonds are formed.

Line 298 FTI=FTIR

As authors answered the gradual intensity loss of DTA curves or FTIR also occurs in single mixture, such as 1: 1 in comparison with pure compound. But this is only due to the decrease of the concentration in the sample of the corresponding substance. It does not mean interaction.

Answer to question 11:

“The increase in the region indicates, therefore, a greater amount of O-H bonds and, therefore, greater interaction of the excipient with the extract.” This region could be indicative of different water content of the sample, thus it cannot be considered as a prove of H-bonding.

 “It corroborates the fact that in the range of 1300 to 1000 cm -1 there is increased C-O stretching”. Authors should explain what does this imply, -C-O stretching increasing is a signal of what kind of interaction ?

Answer to question 13:

I thought that the aim of the work was to choose the excipients that were compatible with the extract, in terms that there are neither physical nor chemical interactions.

Thermograms of samples in which interactions are hypothesized should be incorporated to demonstrate that degradation is effectively retarded.

Author Response

We would like to thank the Reviewer for their valuable comments concerning our work. All issues raised in the review process are addressed in the Revised Manuscript. The corrected text is highlighted in blue (Reviewer's Comment 1). A detailed statement of the modifications made in the manuscript is given below. The text includes more detail to make it more didactic and clearer for the readers.

Round 3

Reviewer 2 Report

Taking into account the aim and content of this work, as well as the no significant relevance of the study about employed polymers, certainly polymers are excipients, I think that this paper is not recommended for “polymers” journal and I strongly suggest that it is resubmitted to a more specific journal such as, “Pharmaceutics” or “Pharmaceuticals”.

Author Response

The editor's comment were accepted and work of adhesion test using pig buccal mucosa was realized, further discussions were inserted to further clarify the subject.  The corrected text is highlighted in yellow. We would like to thank the Editor for their valuable comments concerning our work.